# The PTM profiling of CTCF reveals the regulation of 3D chromatin structure by O-GlcNAcylation

Xiuxiao Tang[1,2,3,4,5,14], Pengguihang Zeng[1,2,3,4,14], Kezhi Liu[1,2,3,4], Li Qing[1,2,3,4], Yifei Sun[1,2,3,4], Xinyi Liu[1,2,3,4], Lizi Lu[1,2,3,4], Chao Wei[1,2,3,4], Jia Wang[6], Shaoshuai Jiang[1,2,3,4], Jun Sun[7,8], Wakam Chang[9], Haopeng Yu[7,8], Hebing Chen ●[10], Jiaguo Zhou[5], Chengfang Xu[11] ✉, Lili Fan ●[12] ✉, Yi-Liang Miao ●[13] ✉ & Junjun Ding ●[1,2,3,4,7] ✉

CCCTC-binding factor (CTCF), a ubiquitously expressed and highly conserved protein, is known to play a critical role in chromatin structure. Post-translational modifications (PTMs) diversify the functions of protein to regulate numerous cellular processes. However, the effects of PTMs on the genome-wide binding of CTCF and the organization of three-dimensional (3D) chromatin structure have not been fully understood. In this study, we uncovered the PTM profiling of CTCF and demonstrated that CTCF can be O-GlcNAcylated and arginine methylated. Functionally, we demonstrated that O-GlcNAcylation inhibits CTCF binding to chromatin. Meanwhile, deficiency of CTCF O-GlcNAcylation results in the disruption of loop domains and the alteration of chromatin loops associated with cellular development. Furthermore, the deficiency of CTCF O-GlcNAcylation increases the expression of developmental genes and negatively regulates maintenance and establishment of stem cell pluripotency. In conclusion, these results provide key insights into the role of PTMs for the 3D chromatin structure.

The mammalian genome is organized into 3D structures ranging from compartments, topological associated domains (TADs), and loops[1–3]. These structures are highly dynamic during development, and their disruption could lead to abnormal gene transcription and diseases[4,5]. CTCF, a ubiquitously expressed and highly conserved DNA-binding protein, plays a critical role in chromatin architecture organization[6–9]. CTCF has been found to form TADs via cohesin-mediated loop extrusion[3,9–12] and the phase separation of CTCF organizes inter-A compartment interactions[13].

The chromatin binding of CTCF is important for chromatin structure organization[14–16]. DNA methylation, RNA binding, and protein–protein interactions have been reported to play roles in CTCF binding[17–22]. However, the mechanisms regulating CTCF binding to chromatin are not fully understood[23–25]. PTMs are essential mechanisms to extend protein functions beyond what is dictated by gene transcription and to regulate cellular physiology[26,27]. Despite the potential diversity of CTCF PTMs[28–30], only a few have been identified, such as acetylation, sumoylation, poly(ADP-ribosyl)nation, and phosphorylation[31–36]. To understand the functional significance of CTCF PTMs, potential PTMs should be systematically mapped from the N to the C terminal. Furthermore, previous studies have found that PTMs of CTCF can affect its chromatin binding in a few loci. For example, the phosphorylation of CTCF decreases its DNA-binding activity in the rRNA gene locus, and mutating CTCF at the major phosphorylation sites leads to the repression of binding to the chicken and human *c-myc* promoters[37,38]. However, the effect of PTMs on the genome-wide binding of CTCF has not been fully elucidated. Meanwhile, the effect

A full list of affiliations appears at the end of the paper. ✉e-mail: xuchengf@mail.sysu.edu.cn; fanlili@jnu.edu.cn; miaoyl@mail.hzau.edu.cn; dingjunj@mail.sysu.edu.cn

of CTCF PTMs on 3D chromatin structure has not been fully understood.

Here, we uncovered the PTM profiling of CTCF in mouse embryonic stem cells (mESCs) and showed that CTCF can undergo O-GlcNAcylation and arginine methylation. To investigate the role of CTCF O-GlcNAcylation, we mapped 3D chromatin architecture, CTCF ChIP-seq, chromatin accessibility, and gene expression in the context of O-GlcNAcylation deficiency. These findings provide information on the importance of CTCF PTMs in organizing 3D genome structure.

## Results

### The PTM profiling of CTCF

It has been reported that CTCF can be phosphorylated, acetylated, sumoylated, and poly (ADP-ribosyl) ated[31–34,39,40]. Many previously reported PTMs were identified by candidate-based approach, a method based on prediction of PTM sites by known databases and followed by validation[38]. In addition to the method above, other PTMs of CTCF have also been identified by the enrichment of interested PTM, such as the enrichment of phosphorylation by antibody or phostag SDS/PAGE[37]. However, these strategies have high levels of inherent biases, which is not conducive to finding new CTCF PTM.

To understand the function of CTCF PTMs, potential PTMs should be mapped from the N to the C terminal with an unbiased method[41]. Hence, we mapped CTCF PTMs unbiasedly and a schematic overview is shown in Fig. 1a. In brief, CTCF was purified from the lysates of WT mESCs and separated by SDS-PAGE (Supplementary Fig. 1a). After coomassie blue staining, the purified protein was analyzed using liquid chromatography tandem MS (LC-MS/MS), followed by analysis of the raw MS files via Byonic database[42]. The PTMs can be classified into four types, among which phosphorylation and acetylation have been found previously (Fig. 1b; Supplementary Fig. 1b)[31,39]. Importantly, we observed CTCF can also be O-GlcNAcylated and arginine methylated, which has not been reported previously (Fig. 1b). To further confirm the results, purified CTCF proteins were enriched with the antibodies of O-GlcNAcylation and arginine methylation, respectively (Fig. 1a). LC–MS/MS analysis revealed one peptide with amino acids 668-672 (TNQPK) as an O-GlcNAcylated peptide (Fig. 1c, d), while four peptides as arginine methylated peptides, which showed a mass shift of 14 or 28 Da on arginine residue (Supplementary Fig. 1c–f).

Taken together, we performed the PTM profiling of CTCF with an unbiased method and identified that CTCF can be O-GlcNAcylated and arginine methylated.

### Validation of O-GlcNAcylation and arginine methylation

O-GlcNAc-specific antibody RL2 and succinylated wheat germ agglutinin (sWGA) have high affinity for O-GlcNAcylated proteins. To further verify the O-GlcNAcylation of CTCF, we purified RL2 and sWGA-bound proteins from mESCs lysates under denatured condition to disrupt protein–protein interactions and probed for CTCF using western blot. Results showed that endogenous CTCF specifically bound to RL2 and sWGA (Fig. 2a), proving CTCF is O-GlcNAcylated. We reconfirmed the O-GlcNAcylation of CTCF by purifying CTCF from mESCs lysates and using western blot with RL2 to recognize O-GlcNAcylated serine and threonine residues (Fig. 2b). Meanwhile, mutation of the CTCF amino acid involved in O-GlcNAcylation showed a decreased signal of RL2 (Fig. 2c). We also removed terminal O-linked glycosidic modifications using β-N-Acetyl-hexosaminidase (β-hex)[43] and the O-GlcNAcylation signal of CTCF was decreased (Fig. 2d). To further confirm the modification, we extracted ion chromatography. Results showed that the peak area of O-GlcNAcylated peptide was 23,340,846,486 and the peak area of unmodified peptide was 15,501,338,529. Therefore, the O-GlcNAcylation exists in 60.09% of CTCF. Since phosphorylation can also occur at serine or threonine residues, we wanted to determine if phosphorylation is present at T668. We performed high-throughput phosphoproteomics and identified approximately 11,000

phosphorylation sites in mESCs. The phosphoproteomic analysis did not detect phosphorylated signal at T668, indicating a low probability of phosphorylation modification occurring at this site. Furthermore, we investigated the distribution of CTCF O-GlcNAcylation across mammalian tissues. We identified a common pattern of modification, in which the CTCF O-GlcNAcylation is found in multiple important tissues (Fig. 2e). Meanwhile, we tested the CTCF O-GlcNAcylation in mouse embryonic fibroblast (MEF), pre-iPSC, which is a late intermediate reprogramming stage, and mESC, the result showed that the signal of CTCF O-GlcNAcylation increases with pluripotency (Fig. 2f).

There are three types of arginine methylation, including monomethylarginine (MMA), asymmetrical dimethylarginine (ADMA), and symmetrical dimethylarginine (SDMA)[44]. To further verify the arginine methylation of CTCF, we used antibodies of ADMA, SDMA, and MMA respectively to purify arginine methylated protein under denatured condition, and probed for CTCF by western blot (Supplementary Fig. 2a–c). We also purified CTCF and performed western blot of ADMA, SDMA, and MMA to reconfirm the results (Supplementary Fig. 2d–f). Moreover, the level of arginine methylation decreased in response to treatment with the arginine methylation inhibitor, periodate oxidized adenosine (AdOx) (Supplementary Fig. 2g). In conclusion, we demonstrated that CTCF possesses three types of arginine methylation.

Taken together, we uncovered the PTMs map of CTCF (Fig. 1b) and demonstrated that CTCF can also be O-GlcNAcylated and arginine methylated.

### The O-GlcNAcylation of CTCF is regulated by OGT

O-GlcNAcylation, an O-β-glycosidic attachment of single N-acetylglucosamine (GlcNAc) to serine and threonine residues, is involved in the regulation of diverse cellular processes[45]. Since in previous studies, O-GlcNAcylation plays important roles in diverse aspects of protein functions, such as protein–protein interaction, protein stability, and enzyme activity[46–48]. We focused on the regulation mechanism and function of CTCF O-GlcNAcylation.

To investigate the regulation mechanism of CTCF O-GlcNAcylation, we focused on the process of O-GlcNAcylation. It is reported that O-GlcNAc transferase (OGT) adds the modification while O-GlcNAcase (OGA) removes it[45]. We confirmed interactions of OGT with CTCF (Fig. 2g). Then we also analyzed the interaction domain of OGT and CTCF by ZDOCK[49] and found the OGT interacts with CTCF in the region of O-GlcNAcylation (Fig. 2h). Furthermore, we knocked down the *Ogt* and found the level of CTCF O-GlcNAcylation was decreased (Fig. 2i).

Taken together, these results show that the O-GlcNAcylation of CTCF is regulated by OGT.

### Deficiency of O-GlcNAcylation enhances chromatin binding of CTCF

To examine the function of the CTCF O-GlcNAcylation, we used CTCF-mAID mESCs as donor cells[14], and transduced the cells with vector encoding WT_CTCF or T668A O-GlcNAc-deficient CTCF (MUT_CTCF), respectively (Fig. 3a). Then, we induced the rapid degradation of endogenously tagged CTCF using 0.5 mM auxin. The clone in which the expression level of exogenous CTCF was at a similar level to that of the endogenous CTCF was collected for further functional experiments (Fig. 3b–d, Supplementary Fig. 3a). After treatment with auxin for 48 h to replace the endogenous CTCF, the expression of core pluripotency factors OCT4, SOX2 and NANOG was similar between WT_CTCF and MUT_CTCF mESCs, and the cells were morphologically identical and were undifferentiated (Supplementary Fig. 3b, c). We used these cells to detect the function of O-GlcNAcylation of CTCF in chromatin binding and 3D chromatin structure.

We firstly investigated the change of CTCF genome-wide binding in response to the mutation of O-GlcNAcylation (Supplementary Fig. 3d). To identify specific differences between chromatin binding of

O-GlcNAc-deficient CTCF (MUT_CTCF) and WT_CTCF, we conducted the spike-in normalized ChIP-seq[50,51], and found that the global chromatin binding of O-GlcNAc-deficient CTCF was increased significantly (Fig. 3e). Consistently, immunoblots showed that chromatin-bound CTCF was increased and non-chromatin-bound CTCF was decreased in the MUT_CTCF mESCs (Fig. 3h). The location distribution of the

enhanced peaks showed that enhanced peaks preferentially bound introns, intergenic regions and insulators (Fig. 3f, Supplementary Fig. 3e). Since the gain of CTCF occupancy is associated with a gain of chromatin accessibility, we performed an assay of transposase-accessible chromatin using high-throughput sequencing (ATAC-seq) experiments. Results showed that chromatin accessibility was

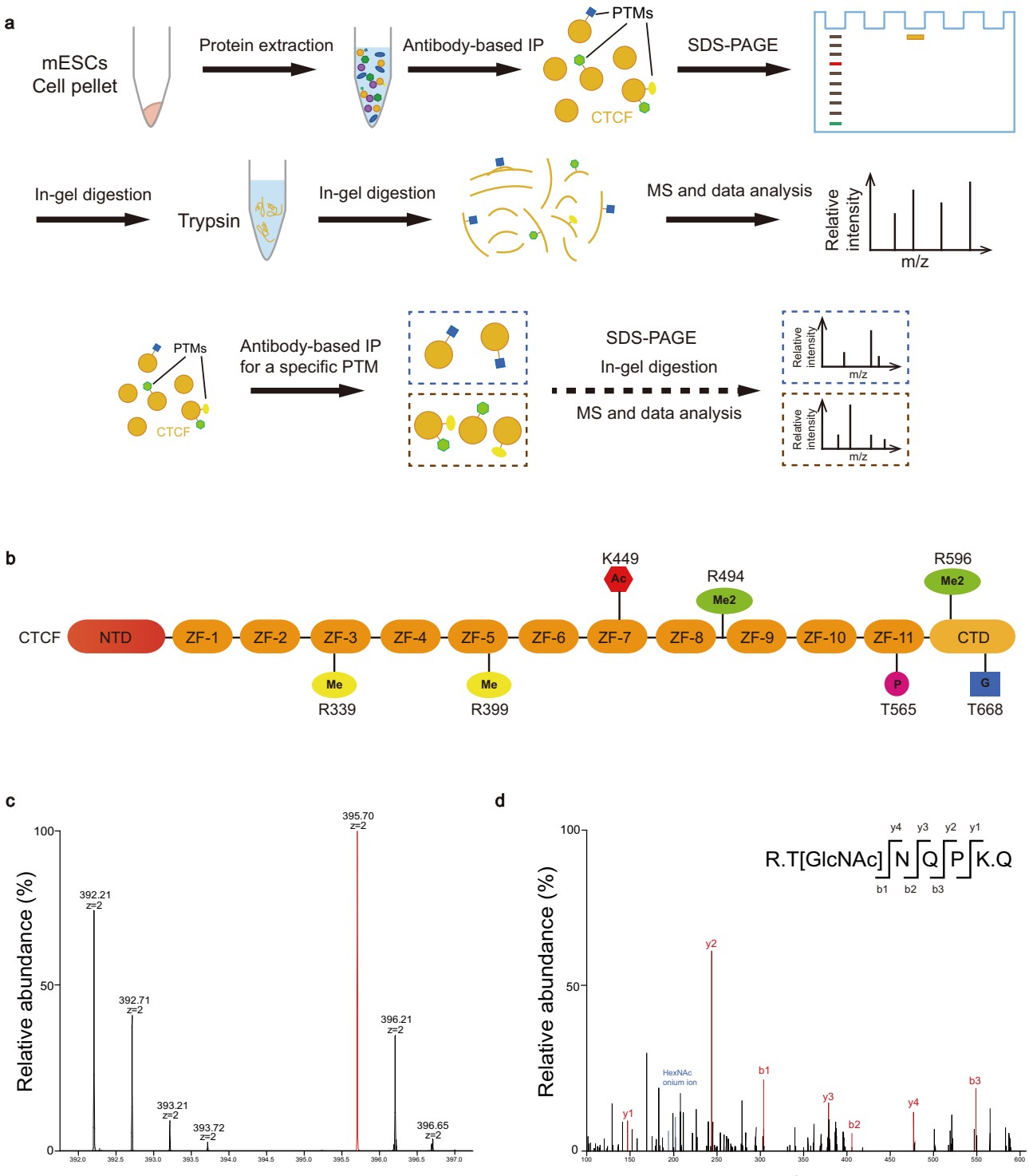

**Fig. 1 | The PTM profiling of CTCF. a** Schematic overview of the identification of PTMs. **b** Illustration of the PTM sites of CTCF identified in mESCs. Me mono-methylarginine, Me2 dimethylarginine, Ac acetylation, P phosphorylation, G O-GlcNAcylation. **c, d**, LC–MS/MS spectra of an O-GlcNAcylated peptide of

endogenous CTCF derived from mESCs and CTCF was purified by immunoprecipitation with antibody. The matched fragment ions were labeled in red. See also Supplementary Fig. 1.

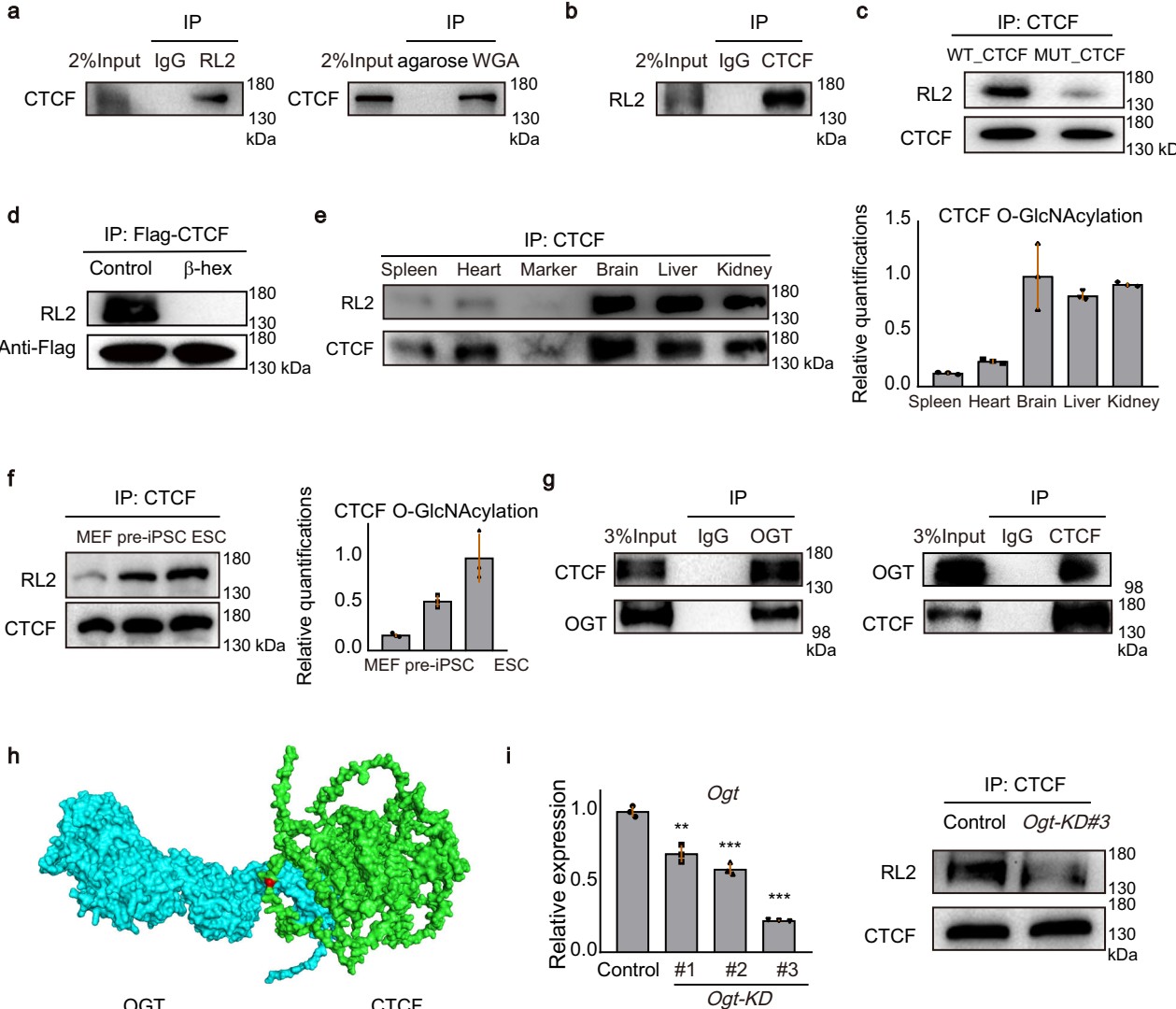

**Fig. 2 | CTCF is O-GlcNAcylated by OGT. a** Whole-cell lysates were denatured and incubated with RL2-conjugated agarose beads (left) or WGA-conjugated agarose beads (right) to enrich O-GlcNAcylated proteins and bound proteins were analyzed by western blot. **b** Whole-cell lysates were denatured and immunoprecipitated with antibody for CTCF from WT mESCs. O-GlcNAcylation of proteins was detected by western blot with O-GlcNAc-specific antibody (RL2). **c** Immunoblot showing the O-GlcNAcylated signal of WT_CTCF and T668A O-GlcNAc-deficient CTCF. **d** The CTCF was purified by immunoprecipitation with Flag antibody in WT_CTCF mESCs and the O-GlcNAcylated signal of CTCF was decreased when treated with β-N-Acetyl-hexosaminidase (β-hex) in vitro. **e** Representative quantifications of CTCF O-GlcNAcylation from different tissues of mouse. Bands were quantified with ImageJ and the bands of the RL2 signal were normalized to purified CTCF to control for equal loading. Data were presented as mean ± SD (*n* = 3 biologically independent samples). **f** Representative quantifications of CTCF O-GlcNAcylation from MEF, pre-iPSC, and mESC. Bands were quantified with ImageJ and the bands of the RL2 signal were normalized to purified CTCF to control for equal loading. Data were presented as mean ± SD (*n* = 3 biologically independent samples). **g** Validation of the endogenous interaction between OGT and CTCF by OGT (left) and CTCF (right) antibody-based IP in mESCs. **h** Predicted protein interaction based on structures of the mouse OGT and CTCF showing that OGT interacted with CTCF in the region of O-GlcNAcylation. OGT was in blue, CTCF was in green and the site of CTCF O-GlcNAcylation was in red. **i** Quantitative PCR analysis of *Ogt* expression after the knockdown of *Ogt* with 3 pairs of siRNA (left) and the signal of CTCF O-GlcNAcylation was decreased (right). This experiment was performed three times, with similar results. Data were presented as mean ± SD (*n* = 3 biologically independent samples). *p*-values by two-tailed *t*-test, significance levels were indicated by asterisks: **\*\*\****p* < 0.01; **\*\*\****p* < 0.001. All western blot experiments were performed three times, with similar results. Source data are provided as a Source Data file.

increased on the regions of enhanced peaks between MUT_CTCF and WT_CTCF (Fig. 3g). Meanwhile, there were more CTCF motifs on the enhanced peaks, which indicates that there could be more CTCF proteins binding to CTCF peaks (Supplementary Fig. 3f). Thereafter, we defined the genes with enhanced peaks located on their promoters (3000 base pairs upstream of gene transcriptional start sites) as enhanced CTCF peaks related genes. Gene Ontology (GO) analysis demonstrated that these genes were highly correlated with developmental process (Fig. 3i, j). To further validate the results, we selected additional clones of WT_CTCF and MUT_CTCF with consistent

expression levels and found that the results were consistent (Supplementary Fig. 3g, h).

Taken together, these results demonstrate that O-GlcNAcylation inhibits CTCF binding to chromatin.

## Deficiency of CTCF O-GlcNAcylation disturbs a subset of loop domains

To measure changes in 3D chromatin structure upon mutation of O-GlcNAcylation, we generated Hi-C profiles of WT_CTCF and MUT_CTCF mESCs (Supplemental Table 4), which revealed high correlation

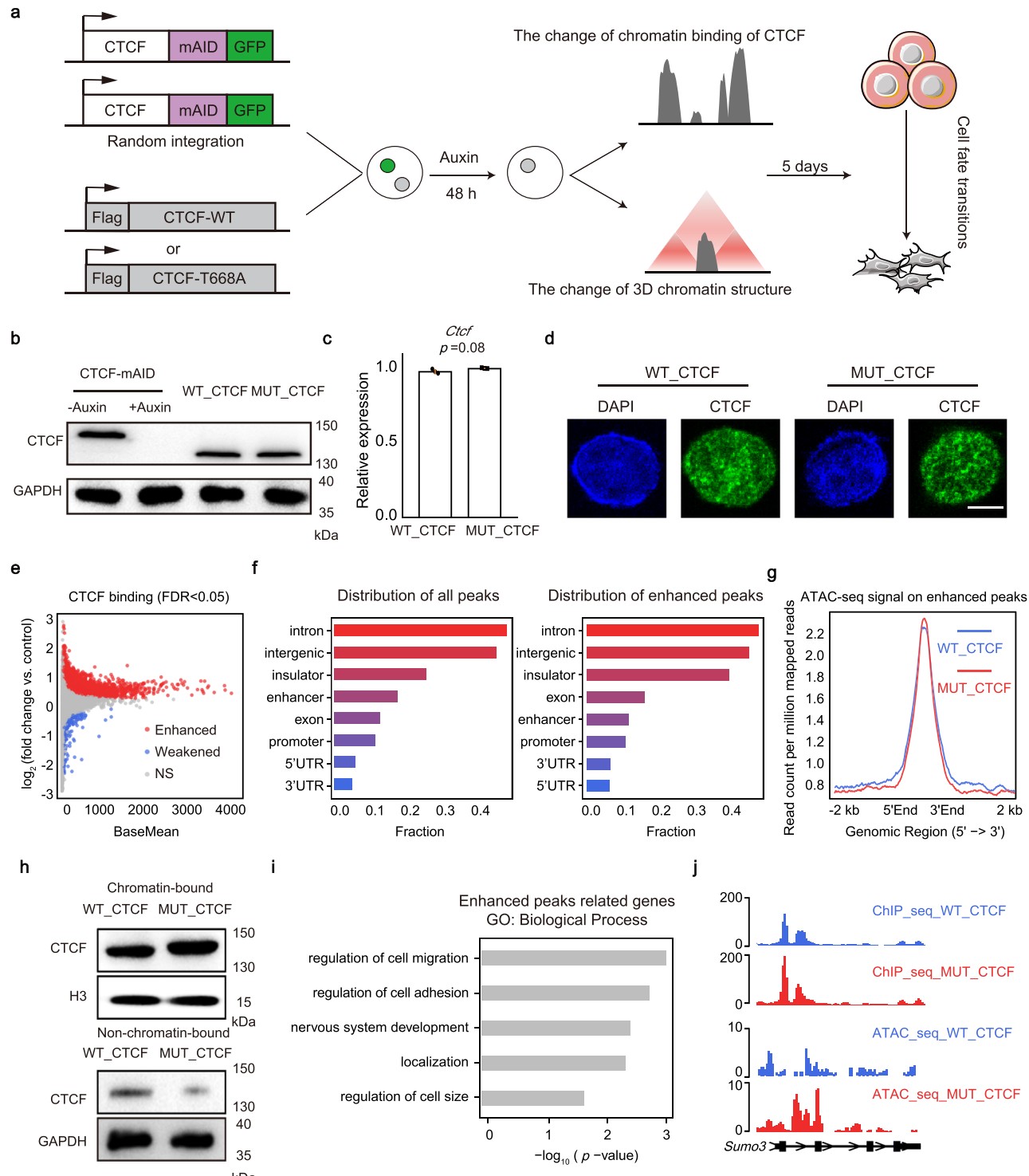

**Fig. 3 | Deficiency of O-GlcNAcylation enhances chromatin binding of CTCF.**
**a** Scheme of the research route and methods in the section on function research.
**b** Western blot of total cell lysates from CTCF-mAID cells before (lane1) or after auxin treatment (lane2), WT_CTCF cells, and MUT_CTCF cells after 48 h auxin treatment to detect the expression of CTCF. GADPH was shown as a loading control. This experiment was performed three times, with similar results. **c** The relative expression of *Ctcf* in WT_CTCF and MUT_CTCF mESCs. Data were presented as mean ± SD (*n* = 3 biologically independent samples). *p*-values by two-tailed *t*-test, *p* = 0.08 compared to control. **d** Immunofluorescence of CTCF in WT_CTCF and MUT_CTCF. Scale bar was 5 µm. This experiment was performed three times, with similar results. **e** MA plot of CTCF ChIP-seq showing global change of CTCF binding

affinity to chromatin in MUT_CTCF mESCs versus WT_CTCF mESCs, FDR < 0.05.
**f** The distribution of all peaks (left) and enhanced peaks of O-GlcNAc-deficient CTCF (right) with indicated genomic features. **g** Profiling of the chromatin accessibility on enhanced peaks. **h** Western blot of chromatin-bound CTCF and non-chromatin-bound CTCF in WT_CTCF and MUT_CTCF mESCs. H3 and GAPDH were shown as loading control. These experiments were performed three times, with similar results. **i** Gene ontology (GO) analysis of enhanced O-GlcNAc-deficient CTCF peaks targeted genes, *p*-values by Fisher's Exact test. **j** Sample of enhanced O-GlcNAc-deficient CTCF peak targeted genes. See also Supplementary Fig. 3. Source data are provided as a Source Data file.

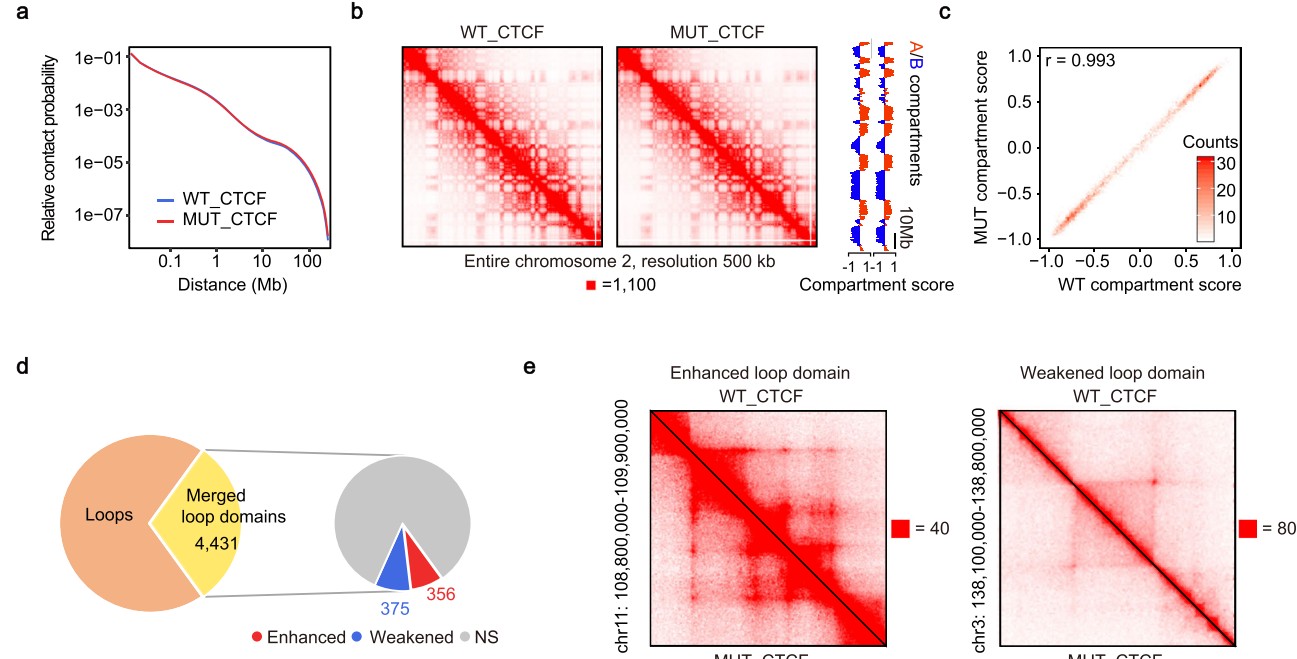

**Fig. 4 | Deficiency of CTCF O-GlcNAcylation disturbs a subset of loop domains.** **a** Overall Hi-C contact probability was not significantly affected. **b** Hi-C contact maps at 500 kb-resolution across entire chromosome 2 in WT_CTCF (left) and in MUT_CTCF (middle). Compartment scores across entirety of chromosome 2 were remarkably consistent (right). **c** Compartment scores were not significantly affected genome-wide, Pearson's r was calculated between WT compartment scores and MUT compartment scores. **d** Number of loops, loop domains, and differential loop domains. **e** Samples of enhanced loop domain (left) and weakened loop domain (right). See also Supplementary Fig. 4. Source data are provided as a Source Data file.

between replicates (Supplementary Fig. 4a). Our WT_CTCF Hi-C showed a high correlation with ESCs in Bonev et al. (Supplementary Fig. 4b) and demonstrated similarity at compartments, TADs and loops[52] (Supplementary Fig. 4d–f). Specifically, 7908 and 7975 TADs, 11,207 and 11,140 loops were identified in WT_CTCF and MUT_CTCF cells respectively, and 7769 TADs and 9931 loops were identified in Bonev's data (Supplementary Fig. 4c).

First, we investigated the extent to which O-GlcNAcylation mutation affects the compartments. The relative contact probability and contact maps did not show a significant change (Fig. 4a, b). The compartment states were maintained upon mutation of O-GlcNAcylation (Fig. 4c). Overall, the mutation of CTCF O-GlcNAcylation does not affect the distribution of A/B compartments. Then, we analyzed the influence on loop domains, which are contact domains whose endpoints form a chromatin loop. Using an approach developed by Rao et al.[3], we identified 3193 loop domains in WT_CTCF and 3263 in MUT_CTCF. Loop domains with both anchors overlapped were merged to generate unique results. In the end, we obtained a total of 4431 merged loop domains, among which 356 were enhanced and 375 were weakened (Fig. 4d, e).

Taken together, these results demonstrate that deficiency of CTCF O-GlcNAcylation disturbs a subset of loop domains.

**Deficiency of CTCF O-GlcNAcylation alters enhancer-promoter interactions associated with cellular development**
To test whether the O-GlcNAcylation of CTCF plays a role in regulating chromatin loops, we defined the loops with CTCF ChIP-seq peaks locating on both anchors as CTCF-related loops and performed the aggregate peak analysis to quantify the differences (Fig. 5a). To examine the types of interactions in each differential loops, we divided CTCF-related loops into enhancer-enhancer (E-E), enhancer-promoter (E-P), promoter-promoter (P-P) and insulator-insulator (I-I) interactions[53]. The O-GlcNAc-deficient CTCF-related loops were preferentially I-I and E-P interactions (Fig. 5b). Since E-P interactions

contribute to gene transcription for cellular development[54], we then analyzed differential CTCF-related E-P interactions. 201 and 316 E-P interactions were strengthened and weakened respectively (Fig. 5c). GO analysis revealed that the genes with promoters overlapping the anchors of strengthened CTCF-related E-P interactions were highly related to developmental processes and the genes with promoters overlapping the anchors of weakened CTCF-related E-P interactions were closely associated with metabolic process and regulation of cell cycle (Fig. 5d). These results indicate deficiency of CTCF O-GlcNAcylation may contribute to cellular development.

Taken together, these results demonstrate that deficiency of CTCF O-GlcNAcylation alters enhancer-promoter interactions which are closely associated with cellular development.

**Deficiency of CTCF O-GlcNAcylation upregulates the expression of developmental genes**
To test whether mutation of CTCF O-GlcNAcylation has any effect on gene expression, we compared the RNA-seq of MUT_CTCF mESCs and WT_CTCF mESCs (Fig. 6, Supplementary Fig. 5a). GO analysis revealed the up-regulated genes were associated with developmental processes, while the down-regulated genes were associated with cell adhesion and metabolic process (Supplementary Fig. 5b, c). These results can partially account for differentiated phenotype of mESCs when culturing the MUT_CTCF mESCs for a long time (Fig. 7). Next, we wondered whether the changes in 3D chromatin structure and CTCF binding have any effect on gene expression. Since deficiency of O-GlcNAcylation enhanced chromatin binding of CTCF, we then want to determine if the strengthened CTCF-related E-P interactions is related to the enhanced chromatin binding of CTCF. We analyzed the reads difference of CTCF peaks on strengthened CTCF-related E-P interactions and found that overall CTCF binding was enhanced (Fig. 6a). Further analysis revealed that 65.52% strengthened CTCF-related E-P interactions had enhanced CTCF peaks on them (Fig. 6b). We then examined the group of genes targeted by strengthened

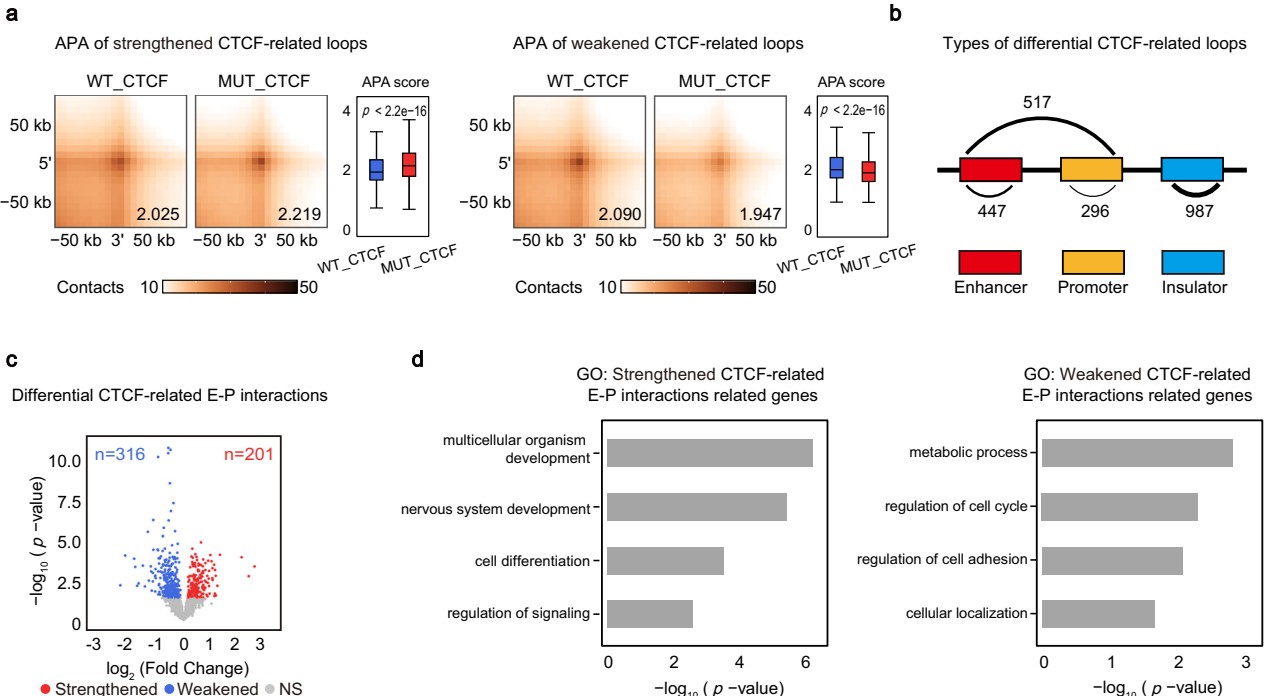

**Fig. 5 | Deficiency of CTCF O-GlcNAcylation alters enhancer-promoter interactions associated with cellular development. a** Aggregate peak analysis (APA) was used to measure the strength of strengthened loops (left) and weakened loops (right). Mean APA enrichment scores were shown on the right bottom APA plot, and comparisons of APA scores were shown in the box plots. Boxplots present the median and 25th and 75th percentile, with the whiskers extending to 1.5 times the interquartile range. Two-tailed *t*-tests were performed, $p < 2.2e-16$ (left), $p < 2.2e-16$ (right). **b** Summaries of the major types of differential CTCF-related loops between MUT_CTCF and WT_CTCF mESCs. **c** Volcano plot comparing differential CTCF-related E-P interactions in MUT_CTCF mESCs against WT_CTCF mESCs. Gray dots, loops with no significant change; blue dots, weakened loops; red dots, strengthened loops, *p*-values by exact test based on over-dispersed Poisson regression, performed by R package diffloop. **d** GO analysis of strengthened CTCF-related E-P interactions related genes (left) and weakened CTCF-related E-P interactions related genes (right), *p*-values by Fisher's Exact test. Source data are provided as a Source Data file.

CTCF-related E-P interactions with enhanced CTCF peaks and found that 63.64% of genes have increased expression (Fig. 6b). GSEA showed that these genes were enriched in MUT_CTCF mESCs, indicating they were prone to be up-regulated in MUT_CTCF mESCs (Fig. 6c). GO enrichment analysis further revealed these genes were enriched for biological processes related to the function of mESCs, including the ability to proliferate indefinitely in vitro (self-renewal) and differentiate into cells from all three germ layers (pluripotency) (Fig. 6d). Specific examples of these genes include *Hox* genes, like *Hoxd8* and *Hoxd9*, which are part of an important gene family involved in limb development and stem cell differentiation[55] (Fig. 6f). GO analysis suggested that mutation of O-GlcNAcylation may affect the ability of proliferation and lead to differentiation of MUT_CTCF mESC cell lines.

Taken together, these results indicate that after the deficiency of O-GlcNAcylation, strengthened CTCF-related E-P interactions with enhanced CTCF peaks upregulate the expression of developmental genes (Fig. 6e).

**Deficiency of CTCF O-GlcNAcylation negatively regulates maintenance and establishment of pluripotency**
Emerging evidence suggests that CTCF plays an important role in cell fate transitions. For example, overexpression of CTCF improves reprogramming[56], knockout of CTCF impedes cell differentiation from ESCs to neural precursor cells (NPCs)[16] and CTCF is a barrier for 2C-like reprogramming[57]. Meanwhile, large number of studies have shown that O-GlcNAcylation plays a role in pluripotency of mESCs. For instance, O-GlcNAcylation of OCT4 and ESRRB facilitates the maintenance of pluripotency[47,58], whereas O-GlcNAcylation of SOX2 inhibits pluripotency[46]. Importantly, we demonstrated above that deficiency of

CTCF O-GlcNAcylation affects chromatin structure associated with cellular development. Hence, we wondered whether O-GlcNAcylation of CTCF is important for cell fate transitions.

To answer this question, WT_CTCF and MUT_CTCF mESCs were cultured for a prolonged period. The cell viability was inhibited in MUT_CTCF cells (Fig. 7a) and MUT_CTCF cells grew slower than WT_CTCF cells due to an elongated G1 phase (Fig. 7b). Moreover, mutation of O-GlcNAcylation resulted in the reduction of total colony numbers with an increased proportion of partially differentiated populations (Fig. 7c, d, g). The expressions of ectodermic marker genes, *Pax3* and *Fgf5*, were increased (Fig. 7e) while the core pluripotency factors, *Oct4* and *Nanog*, were decreased (Fig. 7f). These results indicate O-GlcNAcylation is required for maintenance and self-renewal of mESCs. To study the potential roles of O-GlcNAcylation of CTCF in differentiation, we performed embryoid body (EB) differentiation, which is a spontaneous differentiation of mESCs into cells of all three germ layers. During EBs formation, the size of EBs was larger in MUT_CTCF cells (Fig. 7h) and the expressions of ectodermic markers, *Fgf5* and *Pax3*, were increased in MUT_CTCF cells compared with those in WT_CTCF cells (Fig. 7i). Furthermore, we performed directional differentiation from mESCs to NPCs. As shown in Supplementary Fig. 6a, the expressions of the NPCs markers were decreased, indicating that mutation of O-GlcNAcylation of CTCF inhibits the differentiation into NPCs. At the same time, we explored the potential roles of O-GlcNAcylation of CTCF in establishment of pluripotency during somatic cell reprogramming. We found that mutation of O-GlcNAcylation decreased reprogramming efficiency (Fig. 7j).

Taken together, our data demonstrates that O-GlcNAcylation of CTCF is important for the maintenance and establishment of pluripotency.

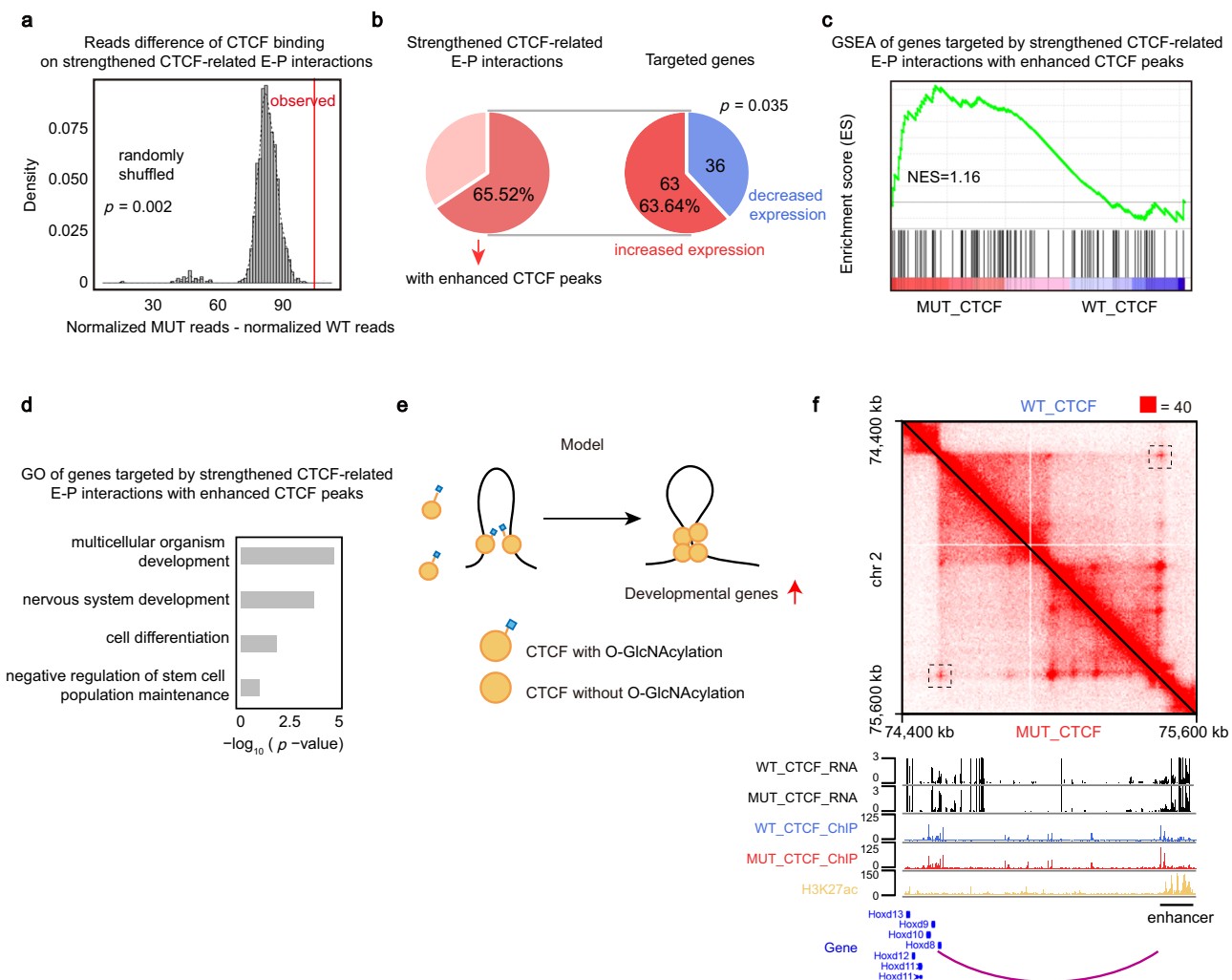

**Fig. 6 | Deficiency of CTCF O-GlcNAcylation upregulates the expression of developmental genes. a** Permutation test showing CTCF had higher level of binding on strengthened CTCF-related E-P interactions than randomly shuffled set of CTCF peaks. *p*-value by one-tailed test based on the assumption of normal distribution, *p* = 0.002. **b** Pie chart showing the percent of strengthened CTCF-related E-P interactions with enhanced CTCF peaks (left) and the percent of increased expression genes targeted by strengthened CTCF-related E-P interactions with enhanced CTCF peaks (right). *p*-value by one-tailed chi-square test,

*p* = 0.035. **c** GSEA of genes targeted by strengthened CTCF-related E-P interactions with enhanced CTCF peaks. **d** GO analysis of genes targeted by strengthened CTCF-related E-P interactions with enhanced CTCF peaks, *p*-values by Fisher's Exact test. **e** Model of the role of O-GlcNAcylation in CTCF chromatin binding and 3D chromatin structure. **f** Representative genomic loci showing up-regulated genes targeted by strengthened E-P interaction with enhanced CTCF peaks. Source data are provided as a Source Data file.

## Discussion

In this study, we performed the PTM profiling of CTCF and demonstrated CTCF can also be O-GlcNAcylated and arginine methylated, providing a basis for further functional studies on the PTMs of CTCF. Functionally, we revealed that O-GlcNAcylation inhibits CTCF binding to chromatin, and deficiency of CTCF O-GlcNAcylation results in the disruption of loop domains and the alteration of chromatin loops associated with cellular development. These findings provide a molecular mechanism for regulation of 3D chromatin structure and indicate the importance of PTMs in chromatin organization. Among the altered loops, some of them are weakened loops. We also briefly discussed the potential mechanisms for the weakening of the loops. Since previous study has indicated the impact of OCT4 on CTCF loops[22], we speculate that it also has an effect on the weakened CTCF loops. To investigate this, we analyzed the binding of OCT4 on weakened CTCF-related E-P interactions. The permutation test results indicated that OCT4's binding was significantly higher, whether it's on the anchors of weakened CTCF-related E-P interactions or the CTCF peaks of weakened CTCF-related E-P interactions (Supplementary Fig. 7a, b). Meanwhile,

we conducted ChIP-qPCR experiments and found that, after the mutation of O-GlcNAcylation, the chromatin binding of OCT4 decreased on the anchor of the weakened loops (Supplementary Fig. 7c, d). However, the binding of OCT4 on the anchor of the strengthened loops remained unchanged (Supplementary Fig. 7c). These results suggest that the weakened loops may be attributed to the reduced binding of OCT4. As for why the mutation of CTCF O-GlcNAcylation leads to a decrease in OCT4 binding on the anchor of weakened loops, we found that the mutation of O-GlcNAcylation weakened the interaction between CTCF and OCT4 (Supplementary Fig. 7e). Therefore, we speculate that at the anchor of weakened loops, the mutation of O-GlcNAcylation weakens the interaction between CTCF and OCT4, leading to a decrease in OCT4's chromatin binding, which may subsequently result in a reduction in the strength of CTCF-related loops.

CTCF is a widely expressed protein with thousands of chromatin-binding sites[6,8,59,60]. Depletion of its zinc finger domain or RNA-binding domain has different effects on the chromatin binding of CTCF and 3D chromatin structure[19,25,61,62]. The PTM profiling performed in this study

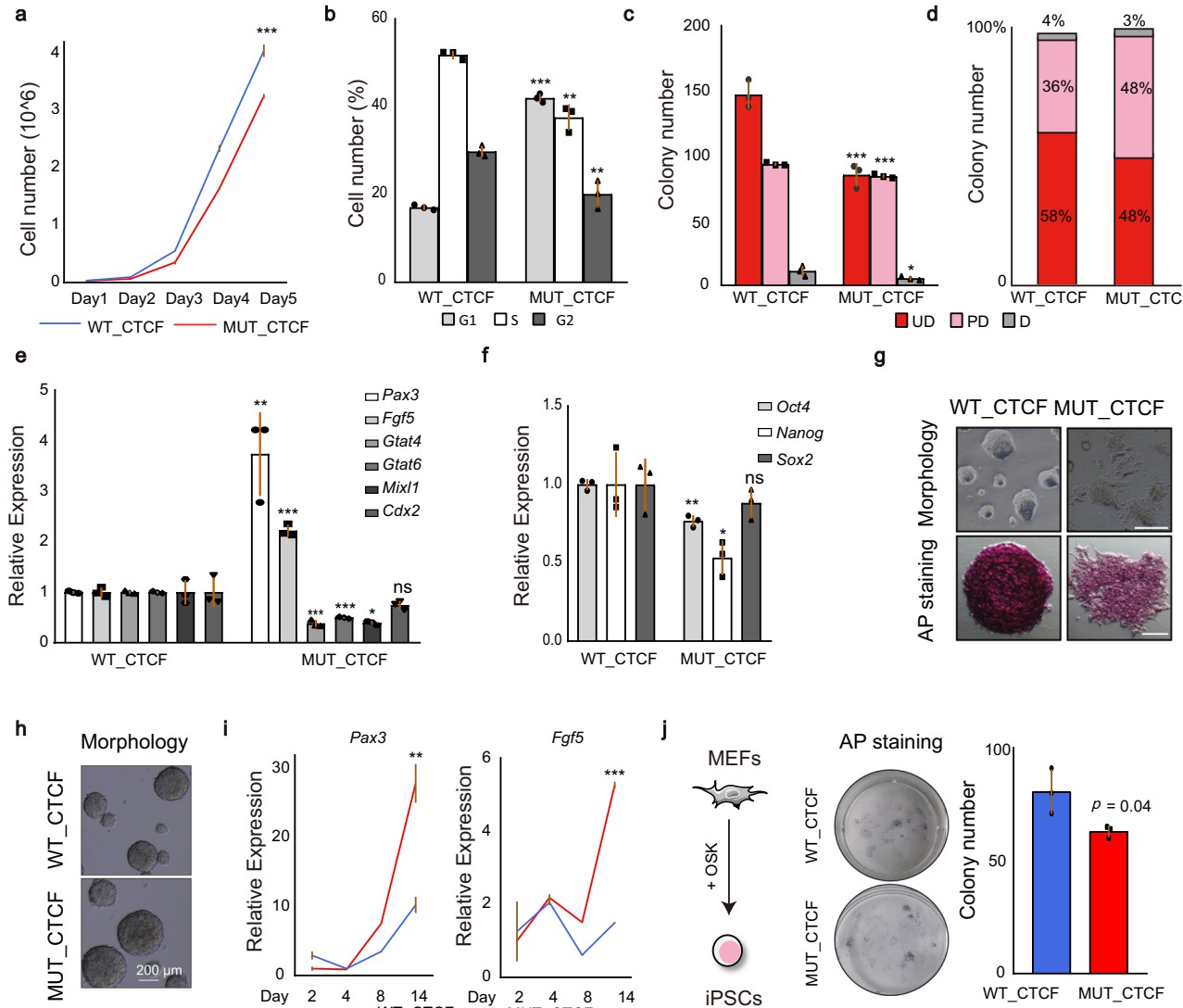

**Fig. 7 | Deficiency of CTCF O-GlcNAcylation negatively regulates maintenance and establishment of pluripotency. a** Growth curve analysis of WT_CTCF and MUT_CTCF cells in 5 days. **b** Cell cycle analysis of WT_CTCF and MUT_CTCF cells in 5 days. **c, d**, Colony formation assay for WT_CTCF and MUT_CTCF cells in 5 days. Colonies were stained for AP activity and divided into three categories (UD uniformly undifferentiated, PD partially differentiated or mixed, and D differentiated) as indicated. **e** Quantitative PCR analysis of lineage markers after mutating CTCF O-GlcNAcylation for 5 days. **f** Quantitative PCR analysis of pluripotency markers after mutating CTCF O-GlcNAcylation for 5 days. **g** Morphology of WT_CTCF and MUT_CTCF cells in 5 days with (bottom) or without (top) AP staining. Scale bar was 200 μm (top) and 50 μm (bottom). This experiment was performed three times, with similar results. **h** Morphology of WT_CTCF and MUT_CTCF cells after EB differentiation. Scale bar was 200 μm. This experiment was performed three times, with similar results. **i** Quantitative PCR analysis of lineage markers during embryoid body (EB) differentiation. **j** Reprogramming efficiency of MUT_CTCF compared with WT_CTCF. AP staining (left) and quantitation (right) of AP positive colony numbers. O OCT4, S SOX2, K KLF4. MEFs mouse embryonic fibroblasts, iPSCs induced pluripotent stem cells. See also Supplementary Fig. 6 and Supplemental Table 2. All statistical tests were two-sided Student $t$-test. Significance levels were indicated by asterisks: $*p < 0.05$; $**p < 0.01$; $***p < 0.001$ (not significant, denoted as n.s.). All data were presented as mean ± SD ($n$ = 3 biologically independent samples). Source data are provided as a Source Data file.

shows that PTMs are widely distributed in various regions (Fig. 1b), and the role of PTMs in different regions is also worth investigating. Meanwhile, CTCF has been reported to favor binding to consensus motif, which is present at 55,000–65,000 sites in the genome, but 30%–60% of CTCF-binding sites show cell-type-specific pattern[7,18,63–65]. Deficiency of O-GlcNAcylation selectively enhances chromatin binding of CTCF in promoter of developmental genes in mESCs (Fig. 3), indicating that PTMs play a role in the selection of CTCF binding. Furthermore, the cell-type-specific CTCF-binding sites result in selective gene expression[66,67]. We found that the level of CTCF O-GlcNAcylation varies across different tissues (Fig. 2), which may be involved in CTCF-regulated gene expression in different cell types.

CTCF plays a key role in the organismal development. Studies have showed that removal of CTCF from oocytes leads to embryo lethality and homozygous null mutant embryos die by the pre-implantation[68–70]. Depletion of CTCF from cardiac progenitors results in lethality in mice[71]. Knockout of CTCF gene from neural cells also causes lethality[72]. It was showed that CTCF O-GlcNAcylation is common among different cell types (Fig. 2), the role of CTCF O-GlcNAcylation in the organismal development remains to be studied. Abnormity of CTCF-mediated chromatin structures leads to disease[5,73], we found that CTCF O-GlcNAcylation regulates 3D chromatin structures, whether the O-GlcNAcylation is related to the abnormal 3D structure is worth studying.

## Methods

### Cell culture

ESC lines R1[74] was used in this study and cultured in gelatin-coated dish with ESC medium, which consisted of DMEM (Hyclone, #SH30022.01), 15% (v/v) fetal bovine serum (FBS, Lonsera, #S712-012S), 0.1 mM β-mercaptoethanol (Sigma, #M6250), 2 mM L-glutamine (Thermo Fisher, #35050061), 0.1 mM nonessential amino acids (Thermo Fisher, #11140050), 1% (v/v) nucleoside mix (Sigma), 1000 U/mL recombinant LIF (Millipore).

Mouse CTCF-EGFP-AID ESCs were presented by Bruneau lab[14] and cultured in gelatin-coated dish with ESC medium. For the depletion of CTCF, the cells were cultured in ESC medium with 0.5 mM auxin (Sigma, #I5148-2g) for 2 days. To establish the T668A glycosylated mutant CTCF (MUT_CTCF) and WT CTCF (WT_CTCF) in CTCF-EGFP-AID ESCs, whose exogenous CTCF was in similar level with endogenous CTCF in CTCF-EGFP-AID ESCs, the fragment of MUT_CTCF and WT_CTCF was subcloned into pJD105[75] and the Fugene HD (Promega, #E2311) was used for transfection and the cells were selected by Hygromycin B (Thermo Fisher, #10687010). Cell clones were picked and then cultured in ESC medium with 0.5 mM auxin for 2 days to degrade endogenous CTCF. The clone whose expression level of exogenous CTCF was similar with endogenous was identified by anti-CTCF antibody and was used in the functional experiments.

To knockdown *Ogt*, we used lipofectamine™2000 for siRNA transfection. Gene expression was detected through RT-qPCR after 72 h. The cells with the highest knockdown efficiency were used for downstream experiments.

### NPC differentiation

The cell line was cultured in N2B27 medium which consisted of a 1:1 mixture of DMEM supplemented with 1× N2 (Gibco, #17502048), NEAA, 1 mM L-glutamine, and 0.1 mM β-mercaptoethanol with Neurobasal (Thermo Fisher, #21103049) supplemented with B27 (Gibco, #17504044). After 7 days, the cells were harvested and the total mRNA was extracted for RT-qPCR.

### Antibodies

Antibodies used in this study were anti-CTCF (Active Motif, #61311), anti-CTCF (Abclonal, #A19588), anti-RL2 (abcam, #ab2739), anti-OGT (GeneTex, # GTX109939), anti-Mono-Methyl Arginine (MMA) antibody (CST, #8015), anti-Symmetric Di-Methyl Arginine (SDMA) antibody (PTMBIO, #PTM-617RM), anti-Asymmetric Di-Methyl Arginine (ADMA) antibody (PTMBIO, #PTM-605RM), anti-Acetyllysine antibody (PTMBIO, #PTM-105RM), anti-GAPDH (Abclonal, #AC001), anti-OCT4 (Santa, #sc-5279), anti-SOX2 (Santa, #sc-36582-3), anti-NANOG (Bethyl, #A300-397A), anti-H3 (Santa, #sc-17576). Goat anti-rabbit-IgG (H + L)−HRP (CST, #7074s), and goat anti-mouse-IgG (H + L)-HRP (CST, #7076s) were used as secondary antibodies for western blotting.

### The purification of CTCF

A total of 50 million cells were harvested by trypsinization, washed with cold PBS, and frozen in liquid nitrogen. Whole-cell pellets were lysed in lysis buffer (50 mM HEPES pH = 7.6, 250 mM NaCl, 0.1% NP-40, 0.2 mM EDTA, 0.2 mM PMSF, 1× protease inhibitor cocktail) on ice. The lysates were cleared by centrifugation at 16,000 × *g* for 10 min at 4 °C and the supernatant was protein extraction. Antibody-based purification was performed to detect the PTMs of endogenous CTCF. Briefly, CTCF antibody was conjugated with Protein G agarose (Roche, #11243233001) by incubating in IP DNP buffer (20 mM HEPES pH = 7.6, 0.2 mM EDTA, 1.5 mM MgCl₂, 100 mM KCl, 20% glycerol, 0.02% NP-40, 1× protease inhibitor cocktail) overnight at 4 °C and washed twice. Then, protein extraction was added to the Protein G agarose and rotated at 4 °C. After 12 h, the supernatant was removed and the antibody-conjugated Protein G agarose was washed twice with IP DNP buffer. 2× SDS loading buffer was added to the antibody-conjugated

Protein G agarose and the protein was eluted by boiling 5 min at 95 °C. SDS-PAGE analysis was applied to assess purification efficiency and the remaining protein was kept in −80 °C for mass spectrometric and western blot.

### Mass spectrometric analysis of CTCF PTMs

Coomassie-stained SDS-PAGE gel band was excised and incubated with 200 μL of 100 mM ammonium bicarbonate with 25% ACN at 37 °C. After 30 min, the supernatant was removed and the gel slices was incubated with 10 mM DTT solution at 60 °C for 30 min. The DTT solution was removed and 100 mM IAA solution was added and incubated sample at 37 °C for 15 min in the dark with shaking. After then, IAA solution was removed and 100 mM ammonium bicarbonate/25% ACN was added to rinse gel slices. In order to digest the gel, 500 μL of ACN was added to shrink gel pieces and was removed after 15 min. 100 mM ammonium bicarbonate was added to recover gel and 1 mg/mL trypsin stock solution was added in the sample to digest overnight at 37 °C. The gel pieces were extracted three times by adding 50 μL of 25% ACN/0.1% TFA solution and incubated at 37 °C for 5−15 min. The supernatant was transferred to a new tube and the peptides were completely dried under vacuum. The peptides were desalted using reverse-phase solid-phase extraction cartridges (Sep-Pak C-18), completely dried under vacuum, and were resuspended in 0.1% formic acid before LC−MS/MS analysis. The mass spectrometry experiments were repeated twice and the coverage of the protein was 68.26% and 65.33%, respectively. Only modifications identified in both mass spectrometry experiments were considered as modifications identified for CTCF.

### Enrichment of O-GlcNAcylated, methylated, and acetylated proteins

25 million cells were individually used for the enrichment of O-GlcNAcylated, methylated, and acetylated proteins and protein extraction was the same as above and denatured by boiling for 10 min at 95 °C. For enrichment of O-GlcNAcylated protein, WGA and RL2 purifications were carried out as described elsewhere[76]. Denatured proteins were incubated overnight at 4 °C with agarose-bound WGA resin (Vector Laboratories, #AL-1023S) or with RL2-conjugated Protein G agarose. The agarose was then washed twice with lysis buffer and the eluted proteins were analyzed by western blot. For enrichment of methylated proteins, anti-Mono-Methyl Arginine (MMA) antibody, anti-Symmetric Di-Methyl Arginine (SDMA) antibody and anti-Asymmetric Di-Methyl Arginine (ADMA) antibody were used respectively as above. For enrichment of acetylated proteins, anti-Acetyllysine antibody was used as above. Agarose without WGA or IgG-conjugated agarose were used as control.

### Identification of phosphorylation modification

$2 \times 10^7$ cells were collected for protein extraction and proteins were digested into peptides. Phosphorylated peptides were enriched and identified by mass spectrometry according to ref. [77].

### The treatment of β-N-Acetyl-hexosaminidase (β-hex), and periodate oxidized adenosine (AdOx)

The treatment of β-N-Acetyl-hexosaminidase (β-hex, NEB, #P0721S) was performed according to the manufacturer's instructions, and 4 μg purified CTCF was used. For the treatment of periodate oxidized adenosine (AdOx, TargetMol, #T22231), mESCs were treated with 30 μM AdOx and DMSO for 36 h and harvested for the purification of CTCF.

### Co-immunoprecipitation

10 million cells were collected and nuclear extracts were prepared from mESCs as described[78]. Endogenous CTCF was immunoprecipitated with 5 μg of CTCF antibody pre-bound to Protein G agarose and co-immunoprecipitated OGT was identified by western blot with the

antibody of OGT. The immunoprecipitation of OGT was done as the same.

## Tissue protein extraction

Tissue protein used in Fig. 2e was generously provided by Dr. Ma from Peking University. The tissue was dissected on ice in PBS/10% FBS. Each tissue was separated and placed to separate tubes using fine-pointed forceps. The dissected tissues were weighed, and then washed with PBS on ice. A ratio of ~1 g of tissue to 20 mL T-PER Reagent (Thermo, #78510) was added to the tissue sample. After homogenizing, the tissues were centrifuged for 5 min at $10,000 \times g$. The supernatant was collected and the protein was quantified by BCA Protein Assay Kit. For the purification of CTCF, each tissue sample weighing 200 mg was used for protein extraction and 500 μg protein was used as total input to purify CTCF.

## Western blot

Samples were electrophoresed on SDS-PAGE gels and transferred to PVDF membranes (BIO-RAD, #1620177), and the membranes were blocked in 5% BSA at room temperature. After 1 h, the membranes were incubated with the primary antibody overnight at 4 °C, washed three times, and incubated with peroxidase-labeled secondary antibody for 1 h at room temperature. After three washes with TBST, bands were visualized with ECL substrate (BIO-RAD, #1705061) and imaged with a CCD camera. All uncropped and unprocessed scans of blots were provided in the Source Data file.

## Immunofluorescence

Cells were grown on gelatin-coated glass for 24 h and then fixed in 4% paraformaldehyde (PFA, Solarbio, #P1110) for 15 min. After washing with PBS, cells were permeabilized with 0.25% Triton X-100 (AMERSCO, #0694-1L) and blocked with 10% bovine serum albumin. After 1 h, cells were incubated with CTCF antibodies in 3% BSA overnight at 2–8 °C. After washing 3 times with PBS, cells were incubated with secondary antibodies for 1 h. The fixed cells were imaged using confocal microscopy.

## ChIP-seq

Cells were crosslinked in 1% formaldehyde (Sigma, #F8775) for 10 min at room temperature and quenched with 125 mM glycine (Sigma, #G7126) for 5 min. After then, the cells were collected and incubated in lysis buffer I (50 mM HEPES-KOH, pH = 7.5, 140 mM NaCl, 1 mM EDTA, 10% glycerol, 0.5% NP-40, 0.25% Triton X-100, protease inhibitors). After 10 min, the cells were collected, resuspended in lysis buffer II (10 mM Tris-HCl, pH = 8.0, 200 mM NaCl, 1 mM EDTA, 0.5 mM EGTA, protease inhibitors), and rotated for 10 min. For sonication, the cells were collected, and resuspended in sonication buffer (20 mM Tris-HCl pH = 8.0, 150 mM NaCl, 2 mM EDTA pH = 8.0, 0.1% SDS, and 1% Triton X-100, protease inhibitors). Sonicated lysates were cleared once by centrifugation at $16,000 \times g$ for 10 min at 4 °C and the supernatant was transferred to 15 ml conical tube. Spike-in Drosophila chromatin (Active Motif, #53083) was added to the supernatant and 50 μL of mixture was saved as input. 10 mg of anti-CTCF (Active Motif, #61311) together with 4 mg spike-in antibody (Active motif, #61686) was added alongside with beads. The remainder of the mixture was incubated with magnetic beads bound with antibody to enrich for DNA fragments overnight at 4 °C. The next day, beads were washed with wash buffer (50 mM HEPES-KOH pH = 7.5, 500 mM LiCl, 1 mM EDTA pH = 8.0, 0.7% Na-Deoxycholate, 1% NP-40) and followed with TE buffer (10 mM Tris-HCl pH 8.0, 1 mM EDTA, 50 mM NaCl). Beads were removed by incubation at 65 °C for 30 min in elution buffer (50 mM Tris-HCl pH = 8.0, 10 mM EDTA, 1% SDS), and supernatant was reverse crosslinked overnight at 65 °C. To purify eluted DNA, 200 μL TE was added to dilute SDS, and 8 μL 10 mg/ml RNase A (Thermo Fisher, #EN0531) was added to degrade RNA. After 2 h, protein was degraded by addition of 4 μL

20 mg/ml proteinase K (Thermo Fisher, #25530049) and incubation at 55 °C for 2 h. Phenol: chloroform: isoamyl alcohol extraction (G-CLONE, #EX0128) was performed followed by an ethanol precipitation. The DNA pellet was then resuspended in 50 μL TE. Library was performed with NEBNext Ultra II DNA library kit (NEB, #E7645). Two biological replicates were performed for each cell line.

## ATAC-seq

Cells were collected, washed with PBS, and incubated in lysis buffer (10 mM Tris-HCl, 10 mM NaCl, 3 mM MgCl2, 0.5% NP-40) for 10 min at 4 °C. After 5 min of centrifugation, TruePrepTM DNA Library Prep Kit V2 for Illumina® (Vazyme, #TD501) was used to make DNA fragmentation. After 30 min, 100 μL VAHTS DNA Clean Beads (Vazyme, #N411) were added to the sample. Then, DNA was collected with a magnet and washed with 80% ethanol. $H_2O$ was added to elute the DNA and library preparation was performed with TruePrepTM Index Kit V2 for Illumina® (Vazyme, #TD202). Libraries were amplified for 12–15 cycles and were size-selected with VAHTS DNA Clean Beads (Vazyme, #N411). Two biological replicates were performed for each cell line.

## In situ Hi-C

In situ Hi-C was performed as in Rao et al.[3] with some modifications. Cells were crosslinked as above. Cells were incubated in lysis buffer (10 mM Tris-HCl pH8.0, 10 mM NaCl, 0.2% Igepal CA630, protease inhibitors cocktail) for 15 min at 4 °C and washed twice. Cells were collected and incubated in 50 μL of 0.5% SDS at 65 °C for 8 min. Then 25 μL 10% Triton-X and 145 μL $H_2O$ were added to quenched SDS. To digest chromatin, 25 μL of 10× NEBuffer2 and 20 μL of MboI (New England Biolabs, #R0147) were added and incubated overnight at 37 °C. The next day, restriction fragments were biotinylated by supplementing the reaction with 37.5 μL biotin-14-dATP (Life Technologies, #19524016), 1.5 μL of 10 mM dCTP (Invitrogen, #18253013), 1.5 μL of 10 mM dGTP (Invitrogen, #18254011), 1.5 μL of 10 mM dTTP (Invitrogen, #18255018), 8 μL of DNA polymerase I, large (Klenow) fragment (New England Biolabs, #M0210) and incubated at 37 °C for 4 h. The end-repaired chromatin was added into 663 μL $H_2O$, 120 μL NEB T4 ligase buffer, 100 μL 10% Triton-X-100, 12 μL 10 mg/mL BSA, 5 μL T4 DNA ligase (New England Biolabs, #M0202) and incubated for 4 h at room temperature. To reverse crosslink, 50 μL of 20 mg/ml proteinase K, 120 μL 10% SDS and 130 μL 5 M NaCl were added in order and incubated at 68 °C overnight. The DNA was precipitated with ethanol and resuspended in 130 μL of Tris-buffer (10 mM Tris-HCl, pH = 8.0) for sonication. Then the liquid volume is replenished to 300 μL. To isolate biotin-labeled ligation junctions, 150 μL of 10 mg/ml Dynabeads MyOne Streptavidin T1 beads (Life technologies, #65602) were washed with 400 μL of 1× Tween washing buffer (5 mM Tris-HCl pH = 7.5, 0.5 mM EDTA, 1 M NaCl, 0.05% Tween 20), collected with a magnet, resuspended in 300 μL of 2× binding buffer (10 mM Tris-HCl pH = 7.5, 1 mM EDTA, 2 M NaCl) and added to the sample. After 45 min, biotinylated DNA was bound to the beads. To remove end-repair and biotin from unligated ends, 88 μL 1× NEB T4 DNA ligase buffer, 2 μL dNTPs, 5 μL T4 PNK (New England Biolabs, #M0201), 4 μL T4 DNA polymerase I (New England Biolabs, #M0203), 1 μL DNA polymerase I, large fragment (Klenow) were added to incubate for 30 min at room temperature. The beads were washed twice in 1× Tween Wash Buffer 2 min at 55 °C. A-tailing was performed by incubating in 90 μL 1× NEB buffer 2, 5 μL dATP, 5 μL Klenow exo at 37 °C for 30 min. The adapter was ligated by incubating in a mixture of 50 μL 1× Quick Ligation Buffer, 2 μL Quick Ligase (New England Biolabs, #M2200), and 3 μL Illumina indexed adapter (New England Biolabs, #E7337) for 15 min at room temperature and followed by adding 2.5 μL User Enzyme. Library was performed with a NEBNext DNA Library Prep Kit and amplified for 10–12 cycles and were size-selected with AMPure XP beads (Beckman Coulter, #A63881). Three biological replicates were performed for each cell line.

### RNA isolation and RNA-seq

Cell pellets were homogenized in RNAzol reagent (MRC, #RN190-500) and processed according to the manufacturer's instructions. Then the total mRNA was extracted for sequencing. Two biological replicates were performed for each cell line.

### Flow cytometry analysis

Single-cell suspensions were prepared and were fixed. The Foxp3/Transcription Factor Staining Buffer Set (eBioscience, #00-5523-00) was used according to the manufacturer's instructions to detect the expression of CTCF. For cell cycle analysis, two days after treatment of auxin, 1000 WT_CTCF, and MUT_CTCF mESCs were seeded into individual wells of a six-well plate. After 4 days, the cells were collected, fixed in 75% ethanol, and washed three times. DAPI (Sigma, #10236276001) was used for staining and cells were analyzed by flow cytometry. Experiments were conducted in three independent triplicates.

### Cell viability assay

The cell counting kit 8 (CCK8) (DOJINDO, #CK04) was used according to the manufacturer's instructions. Experiments were conducted in three independent triplicates.

### Colony formation assay

Two days after treatment of auxin, 1000 WT_CTCF and MUT_CTCF mESCs were seeded into individual wells of a six-well plate. After 5 days, the colonies were stained by alkaline phosphate (AP, Yeasen, #40749ES60). Colonies of undifferentiated cells (UD), partially differentiated cells (PD), and differentiated cells (D) in each well were counted. Experiments were conducted in three independent triplicates.

### EB differentiation assay

$10^5$ WT_CTCF and MUT_CTCF were cultured in standard ESC medium without LIF. Images were taken at day 2, 4, 8, and 14.

### Prediction of protein–protein interaction

Predicted protein structures of CTCF (Q61164) and OGT (Q8CGY8) are obtained from Alphafold Protein Structure Database. Protein–protein interactions are predicted by ZDOCK Server[49] (https://zdock.umassmed.edu/) and visualized by PyMol.

### Quality control of sequencing reads

All the Illumina sequencing reads used in the study were firstly quality controlled by Trim Galore. In detail, we removed the bases with quality below 20 and the adapter sequences from the 3′ end, and filtered the reads with length less than 50 nt.

### RNA-seq data analysis

RNA-seq reads were aligned to mm9 reference genome using STAR[79] with default parameters. The uniquely mapped reads were counted with HTSeq-count. We detected the differentially expressed genes using edgeR[80]. Genes were considered differentially expressed when the $p$-value < 0.05 and the fold change is above 2.0. GSEA analysis was performed by GSEA_Linux_4.0.3[81].

### ChIP-seq and ATAC-seq data analysis

ChIP-seq and ATAC-seq reads were aligned to mm9 reference genome using Bowtie2[82] with default parameters, followed by removing the multiple aligned reads, PCR duplications with SAMtools. Alignment track bigwig files were generated using deepTools[83]. ChIP-seq and ATAC-seq profiling plot were generated by deepTools[83].

ChIP-seq data were aligned to BDGP6 reference genome, followed by removing the multiple aligned reads. Spike-in reads were counted and scale-factors between samples were determined for downstream analyses. For differential binding analysis, we first called CTCF peak by macs2 peak calling pipeline (https://github.com/macs3-project/MACS/wiki/Advanced:-Call-peaks-using-MACS2-subcommands) with multiplication of scale factors to generate bedGraph files. We then concatenated WT_CTCF and MUT_CTCF CTCF peaks to get full set of CTCF peaks, and then used bedtools multicov for reads counting on all CTCF peaks. Read counts were then normalized by scale factors and R package DESeq2 was used to perform differential analysis.

### In situ Hi-C data analysis

Hi-C reads were processed using Hi-C-Pro[84] pipeline: reads were aligned to mm9 reference genome using bowtie2, reads with mapping quality >10 were assigned to MboI restriction fragments, and interaction pairs were reconstructed. Singleton or multi-hits pairs were filtered out, followed by removal of failed ligation products (dangling end pairs, re-ligation pairs, self-cycle pairs) and pairs not able to reconstruct the ligation product. Remaining pairs were then de-duplicated and used for building contact matrices. Contact matrices were normalized by iterative correction and eigenvector decomposition (ICE).

Compartments were identified by CscoreTool[85] on 500 kb-resolution contact matrices. Loop domains were identified using the method proposed by Rao et al. [86]. Loops were annotated by HiCCUPS with default parameters (-r 5000, 10000, 25000 –f 0.1). Differential loops were identified by diffloop[87]. CTCF-related loops were defined as loops with CTCF ChIP-seq peaks located on both anchors. Aggregate peak analysis about loops were performed by GENOVA[88]. The correlations of contact matrices were analyzed by HiCRep[89].

### Definition of regulatory regions

Enhancers were defined as H3K27ac peaks that did not overlap with a promoter. Insulators were defined according to reference[53], and insulators were identified as the subset of CTCF ChIP-seq peaks that overlapped SMC1 ChIA-PET anchors.

### Genomic feature analysis

BEDTools[90] was used to perform analysis about genomic intervals, including intersecting, expanding, flanking and randomly shuffling intervals.

### Gene ontology analysis

Gene symbols were first converted to EntrzID with R package BiomaRt (version 2.42.0)[91]. Gene ontology analysis is based on reference[92].

### Reporting summary

Further information on research design is available in the Nature Portfolio Reporting Summary linked to this article.

## Data availability

Previously published raw reads of ChIP-seq data were downloaded from GSE29218 (CTCF)[1], GSE85185 (CTCF)[14], GSM2417096 (H3K27ac)[93] and re-processed as described in methods. Previously published raw reads of Hi-C data were downloaded from GSE96107[52], and re-processed as described in methods. All datasets are available in GEO under the accession number GSE255897. Other information needed is available form corresponding author upon request. Source data are provided with this paper.

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

## Acknowledgements

We thank the members of the Ding laboratory for discussion, technical advice, and support. We thank Dr. Ma from Peking University for sharing tissue protein. This research was funded by grants from the National Key R&D Program of China (2023YFA1800900), National Natural Science Foundation of China (31970811 and 32170798), the Guangdong Basic and Applied Basic Research Foundation (2021B1515120063) to J.D., The National Natural Science Foundation of China (32100497), the Natural Science Foundation of Guangdong Province, China (2023A1515010197) to C.W., the Natural Science Foundation of Guangdong Province, China (2021A1515010938, 2023A1515010148), the Fundamental Research Funds for the Central Universities, Sun Yat-sen University (23PTPY88) to J.S., the Science and Technology Development Fund, Macau SAR (file 0077/2020/A2) and from the University of Macau (file: MYRG2022-00251-FHS) to W.C.

## Author contributions

X.T. conceived the experiments. X.T., K.L., L.Q., Y.S., L.L. conducted experiments. P.Z., X.T. performed the bioinformatics analysis. X.T. and P.Z. wrote the paper with input from all other authors. X.L., C.W., J.W., S.J., J.S., W.C., J.Z. revised the manuscript. H.Y., H.C. provided guidance in bioinformatics analysis. J.D., Y.M., L.F. and C.X. supervised the project.

## Competing interests

The authors declare no competing interests.

## Additional information

[1]RNA Biomedical Institute, Sun Yat-Sen Memorial Hospital, Zhongshan School of Medicine, Sun Yat-Sen University, Guangzhou 510080, China. [2]Department of Rehabilitation Medicine, The Seventh Affiliated Hospital, Sun Yat-Sen University, Shenzhen, Guangdong 518107, China. [3]Advanced Medical Technology Center, The First Affiliated Hospital, Zhongshan School of Medicine, Sun Yat-Sen University, Guangzhou, Guangdong, China. [4]Center for Stem Cell Biology and Tissue Engineering, Key Laboratory for Stem Cells and Tissue Engineering, Ministry of Education, Sun Yat-Sen University, Guangzhou 510080, China. [5]Department of Pharmacology and Cardiac & Cerebral Vascular Research Center, Zhongshan School of Medicine, Sun Yat-Sen University, Guangzhou 510080, China. [6]GMU-GIBH Joint School of Life Sciences, Guangzhou Medical University, Guangzhou 511436, China. [7]West China Biomedical Big Data Center, West China Hospital/West China School of Medicine, Sichuan University, Chengdu 610041, China. [8]Med-X Center for Informatics, Sichuan University, Chengdu 610041, China. [9]Department of Biomedical Sciences, Faculty of Health Sciences, University of Macau, Taipa, Macau, China. [10]Institute of Health Service and Transfusion Medicine, Beijing 100850, China. [11]The obstetric and gynecology Department of The third affiliated hospital of Sun Yat-Sen University, Guangzhou, China. [12]Guangzhou Key Laboratory of Formula-Pattern of Traditional Chinese Medicine, School of Traditional Chinese Medicine, Jinan University, Guangzhou, Guangdong, China. [13]Institute of Stem Cell and Regenerative Biology, College of Animal Science and Veterinary Medicine, Huazhong Agricultural University, Wuhan, Hubei, China. [14]These authors contributed equally: Xiuxiao Tang, Pengguihang Zeng. ✉e-mail: xuchengf@mail.sysu.edu.cn; fanlili@jnu.edu.cn; miaoyl@mail.hzau.edu.cn; dingjunj@mail.sysu.edu.cn

