## [Peer Review File · Nature Communications]

The PTM profiling of CTCF reveals the regulation of 3D chromatin structure by O-GlcNAcylationEditorial Note: This manuscript has been previously reviewed at another journal. This document only contains reviewer comments and rebuttal letters for versions considered at *Nature Communications*.

REVIEWERS' COMMENTS

Reviewer #1 (Remarks to the Author):

The authors have addressed all my previous concerns, which were all due to the low quality of the Hi-C data. The authors have performed additional Hi-C experiments and sequenced the libraries to a depth of approximately 4 billion contacts. This has resulted in a dramatic increase in the quality of all heatmaps shown in the manuscript. The authors have replaced all figures with new ones obtained with the improved Hi-C datasets. With the additional data analyses and results, the manuscript now describes an interesting finding to explain how CTCF function can be regulated by covalent modifications. The manuscript is appropriate for publication in its current form

Reviewer #2 (Remarks to the Author):

The revised manuscript addresses my comments from the previous submission and I have no further comments.

Reviewer #1 (Remarks to the Author):

The authors have addressed all my previous concerns, which were all due to the low quality of the Hi-C data. The authors have performed additional Hi-C experiments and sequenced the libraries to a depth of approximately 4 billion contacts. This has resulted in a dramatic increase in the quality of all heatmaps shown in the manuscript. The authors have replaced all figures with new ones obtained with the improved Hi-C datasets. With the additional data analyses and results, the manuscript now describes an interesting finding to explain how CTCF function can be regulated by covalent modifications. The manuscript is appropriate for publication in its current form.

Response: Thank you for the review and recognition of our work.

Reviewer #2 (Remarks to the Author):

The revised manuscript addresses my comments from the previous submission and I have no further comments.

Response: We appreciate your evaluation.